# Extra-pair paternity drives plumage colour elaboration in male passerines

**Emma Thibault[1], Sean M. Mahoney[1], James V. Briskie[2], Mateen Shaikh[3], Matthew W. Reudink[1]***

**1** Department of Biological Sciences, Thompson Rivers University, Kamloops, British Columbia, Canada, **2** School of Biological Sciences, University of Canterbury, Christchurch, New Zealand, **3** Department of Mathematics & Statistics, Thompson Rivers University, Kamloops, British Columbia, Canada

* mreudink@tru.ca

## Abstract

The elaborate ornamental plumage displayed by birds has largely been attributed to sexual selection, whereby the greater success of ornamented males in attaining mates drives a rapid elaboration of those ornaments. Indeed, plumage elaboration tends to be greatest in species with a high variance in reproductive success such as polygynous mating systems. Even among socially monogamous species, many males are extremely colourful. In their now-classic study, Møller and Birkhead (1994) suggested that increased variance in reproductive success afforded by extra-pair paternity should intensify sexual selection pressure and thus an elaboration of male plumage and sexual dichromatism, but the relatively few measures of extra-pair paternity at the time prevented a rigorous test of this hypothesis. In the nearly three decades since that paper's publication, hundreds of studies have been published on rates of extra-pair paternity and more objective measures of plumage colouration have been developed, allowing for a large-scale comparative test of Møller and Birkhead's (1994) hypothesis. Using an analysis of 186 socially monogamous passerine species with estimates of extra-pair paternity, our phylogenetically controlled analysis confirms Møller and Birkhead's (1994) early work, demonstrating that rates of extra-pair paternity are positively associated with male, but not female, colouration and with the extent of sexual dichromatism. Plumage evolution is complex and multifaceted, driven by phylogenetic, ecological, and social factors, but our analysis confirms a key role of extra-pair mate choice in driving the evolution of ornamental traits.

## Introduction

Plumage colouration in birds can serve a variety of functions, from crypsis and thermoregulation, to inter- and intra-specific signaling, and is extraordinarily diverse, spanning the entire avian visual colour spectrum from ultraviolet to red [1]. A vast literature supports Darwin's [2] proposition that females should prefer more ornamented males as sexual partners, leading to the elaboration of ornamental traits. Whether a preference for colourful mates arises from pre-existing sensory bias [3–6], or those traits act as honest signals of individual condition or quality [7–9], the expression of ornamental traits and correlated female preference can result in rapid elaboration and a Fisherian runaway process [10].

**Data Availability Statement:** Please note that in our original submission, we indicated that the data would be made available on datadryad.org. However, since all data we used are publicly available, we believe it would be more appropriate

to link directly to the original sources: Dale, James et al. (2016), Data from: The effects of life history and sexual selection on male and female plumage colouration, Dryad, Dataset, https://doi.org/10.5061/dryad.1rp0s Brouwer L, Griffith SC (2019) Extra-pair paternity in birds. Mol Ecol 28:4864–4882. Table S1 https://doi.org/10.1111/mec.15259 In addition, we have added the code for our analyses to the supplemental material.

**Funding:** Funding for this research was provided by an Natural Sciences and Engineering Research Council of Canada (NSERC) Discovery Grant #4603-2018 to MWR and a Thompson Rivers University CUEF UREAP award to ET. The funders had no role in study design, data collection and analysis, decision to publish, or preparation of the manuscript.

**Competing interests:** The authors have declared that no competing interests exist.

Brilliant ornamental colouration is thought to arise most frequently in taxa with strong sexual selection pressures. For example, polygynous, lekking species such manakins and birds of paradise exhibit extremely high reproductive skew and express some of the most elaborate ornaments on earth [4, 11, 12]. However, even within socially monogamous species, extra-pair paternity (EPP) can increase reproductive skew, and lead to an increase the strength of sexual selection [13]. While EPP is extremely common in birds (approximately 90% of socially monogamous bird species mate outside their pair-bond; [14]), rates vary among species, from virtually absent to extremely widespread, with some species having up to 90% of nests containing extra-pair young [14]. Thus, if sexual selection pressures lead to the elaboration of plumage colouration, rates of EPP in socially monogamous bird species should be positively correlated with plumage elaboration and the extent of sexual [15–17].

In a 1994 study, Møller and Birkhead [15] addressed this question and found a direct link between rates of EPP and both plumage colouration and degree of sexual dichromatism. However, paternity analysis was relatively new at that time and relatively few studies were available on rates of EPP. Over the 27 years since Møller and Birkhead [15] published their study using data from 55 species (36 of whom were socially monogamous), hundreds of studies have been conducted on EPP in birds. Recently, Brouwer and Griffith [18] conducted a meta-analysis on all published studies of paternity patterns in birds, examining relationships between EPP ecological and life history characteristics. This comprehensive dataset comprises over 500 studies in over 300 bird species and provides a remarkable resource for comparative studies examining the causes and consequences of variation in EPP.

In addition to a limited dataset of rates of EPP in birds, Møller and Birkhead [15] also used a colour score ranked by human observers rather than a quantifiable technique. Although scores of plumage colour by human observers can provide a reasonable estimate of differences among species, recent widespread interest in examining large-scale evolutionary patterns in the evolution of plumage colouration led Dale et al. [11] to develop a quantitative measure of plumage elaboration for all passerine birds, specifically for use in phylogenetic comparative studies. This metric, derived via analysis of RGB values extracted from illustrations in Birds of the World [19] provides an objective and systematic measure of plumage elaboration.

Here, we revisit Møller and Birkhead's [15] conclusions that plumage elaboration and sexual dichromatism is directly linked to rates of EPP by utilizing a greatly increased sample size of rates of EPP in socially monogamous species (compiled by [18]) and an objective measure of the elaboration of plumage colouration (from [11]). Using a phylogenetically-controlled analysis of 186 socially monogamous passerine species, we sought to determine if, consistent with the earlier findings of Møller and Birkhead [15], EPP is positively correlated with the elaboration of plumage colouration.

## Methods

### Data collection

We extracted rates of EPP (% extra-pair offspring in the population) for 186 species of socially monogamous passerine birds from a comprehensive dataset of published studies by [18]. For species with multiple published studies of extra-pair paternity, we calculated average rates of extra-pair paternity across studies.

To quantify variation in plumage colouration across the 186 passerine species, we utilized colour scores generated by [11]. Though reflectance spectrometry can provide high precision in differences among individuals, populations, and species, there are several challenges to the approach, including attaining study skins from each species for colour analysis, intraspecific variation among populations, and choosing the correct colour patch for analysis. To deal with

these challenges [11], developed a system specifically for use in large-scale phylogenetic analyses that generated red, green, blue (RGB) values from images of the crown, forehead, nape, throat, upper breast, and lower breast of each of the 5,983 species listed in the Birds of the World [19]. From these data [11], calculated the mean RGB values from 6 plumage patches to generate a single metric that describes the overall color of a given species' plumage (for more detailed methods see [11], Methods: Plumage Scores). Higher color scores for both sexes represent species that are more colorful. Importantly [11], verified their colour scores were consistent with estimates from spectrometry on museum specimens ($R^2 = 0.67$, P<0.0001, [11], see their Extended Data Fig 1 and Extended Data "Plumage scores validation analysis" section). [11] used these scores to explore large-scale questions on correlated evolution between sexes and the effects of morphological, social, and life-history traits in the evolution of colour elaboration. As the authors conclude, these colour scores are ideal for hypothesis testing on the function and evolution of colour ornamentation in both males and females—just as we do here.

## Phylogenetic analysis

To control for the effects of shared ancestry among species, we downloaded 1000 possible phylogenies from BirdTree.org (Source of trees: Hackett All Species: a set of 10000 trees with 9993 OTUs each and Ericson All Species: a set of 10000 trees with 9993 OTUs each) to create a phylogeny of all 186 species for which we had paternity and colour data. Analyses were conducted on both backbones but only the results from the Hackett backbone are shown as the results from each backbone are very similar to each other.

## Statistical analysis

All analyses were performed in R 3.5.3 [20] using the *phytools* package [21] and phylogenetically-controlled least squares (PGLS) models in the *nlme* package [22]. We used PGLS analyses to examine whether rates of EPP predicted male and female colour scores and the degree of sexual dichromatism. To calculate sexual dichromatism, we subtracted female colour scores from male colour scores. In addition, because dichromatism can also arise from female elaboration (or loss of male colouration), we also calculated the absolute difference of sexual dichromatism between male and female colour scores. Because [18] recently found that latitude was an important predictor of rates of extra-pair paternity, we also included latitude of the centroid of the breeding range as a co-variate in our PGLS. The analysis was performed on each of the 1000 trees with results agglomerated into posterior distributions of slopes, giving equal weight to each of the 1000 trees. This analysis was conducted separately for both the Hackett and Ericson backbones.

## Results

Rates of extra-pair paternity were positively associated with male plumage colouration (credible interval for the slope: Ericson = 0.022–0.345, Hackett = 0.036–342) (Fig 1A), sexual dichromatism (credible interval for the slope: Ericson = 0.012–0.354; Hackett = 0.034–0.355) (Fig 1C), and the absolute value of sexual dichromatism: (credible interval for the slope: Ericson = 0.043–0.343, Hackett = 0.048–0.335) (Fig 1D). In contrast, rates of extra-pair paternity were not associated with female plumage colouration (credible interval: Ericson = -0.108–0.099, Hackett = -0.110–0.944) (Fig 1B).

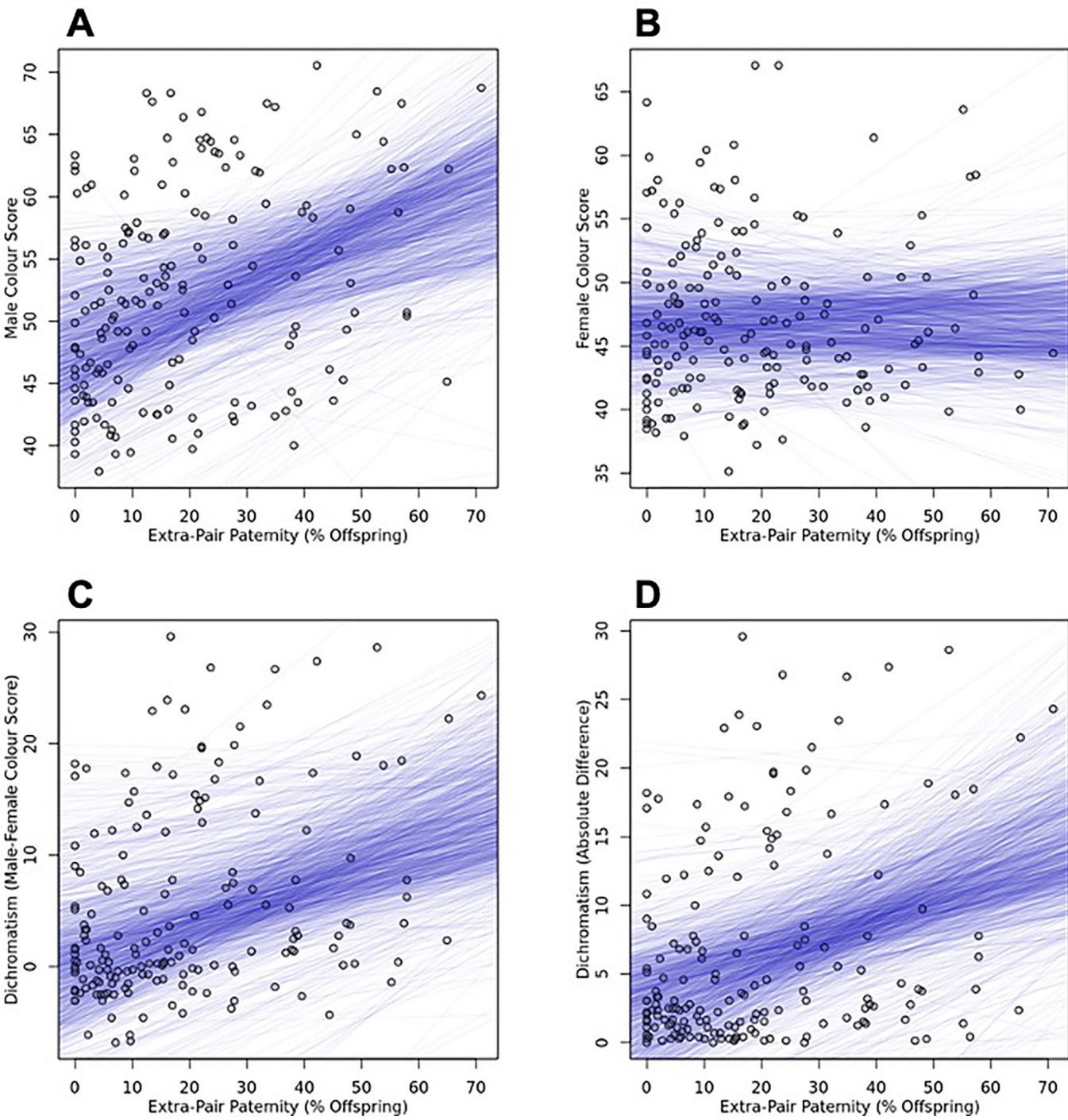

**Fig 1.** Rates of extra-pair paternity are positively associated with male colour scores (A), sexual dichromatism (C), and the absolute value of sexual dichromatism (D), but not female colour scores (B). Each figure shows the posterior distribution of GLS lines from the 1000 GLS models corresponding to the Hackett trees.

## Discussion

In species with high reproductive skew, ornaments can evolve rapidly when those traits are associated with variance in reproductive success. Large-scale comparative studies clearly

demonstrate that mating systems with strong sexual selection pressures (e.g., lek polygyny) have the highest degree of ornamentation [11, 12]. Though reproductive skew in socially monogamous species is relatively low, variance in mating success is increased through EPP [14, 18]. As such, rates of EPP are often used as a proxy for sexual selection pressure [14, 18]. Consistent with sexual selection theory, we should observe a higher degree of male ornamentation in species with greater rates of EPP, as originally demonstrated by Møller and Birkhead [15]. Using an expanded dataset containing rates of EPP in 186 socially monogamous passerine species, we confirm that rates of EPP are positively associated with elaboration in male, but not female, plumage colouration, as well as the degree of sexual dichromatism.

Though our study confirms early work demonstrating that that EPP is linked to plumage elaboration and dichromatism in birds [15], the evolution of plumage colouration in complex and multi-faceted, driven also by the interplay among phylogenetic, ecological, behavioural, and geographic factors [11, 12, 23, 24]. Similarly, EPP in birds is highly variable and driven by a broad range of factors [14, 17, 25], but recent work suggests that broad-scale ecological drivers are poor predictors of EPP and the broad variance in EPP is better explained by ecological and life history factors operating at a finer scale (i.e., among groups of species or populations of a species; [18]). One limitation to our study is that we examined rates of EPP and colouration at the species level. However, given the high variation in EPP and colouration across populations, future studies using standardized spectrophotometric analyses at the population level may provide more insight into the evolutionary processes driving plumage elaboration.

One of the challenges in understanding the factors that influence the evolution of sexual dichromatism is determining whether dichromatism evolved through a loss of ornamentation in females or a gain of ornamentation in males, both of which appear to have occurred repeatedly across taxa [26–29; reviewed in 30]. These gains and losses are often linked to ecological pressures such as, for example, migration [29] and latitude [31]. Thus, while EPP may be an important driver of plumage evolution in migratory birds, future work would benefit from examining the relative strength and role of EPP amidst the other multitudinous ecological and life history factors that shape colouration.[27,28]

## Supporting information

**S1 File.**
(HTML)

**S2 File.**
(HTML)

## Acknowledgments

We thank Dale et al. (2015) and Brouwer and Griffith (2019) for creating, and making publicly available, their excellent datasets on colour and extra-pair paternity, respectively. We also thank Dr. Nancy Flood for her assistance and feedback on this study.

## Author Contributions

**Conceptualization:** Sean M. Mahoney, James V. Briskie, Matthew W. Reudink.

**Data curation:** Emma Thibault, Mateen Shaikh, Matthew W. Reudink.

**Formal analysis:** Emma Thibault, Sean M. Mahoney, Mateen Shaikh, Matthew W. Reudink.

**Funding acquisition:** Emma Thibault, Matthew W. Reudink.

**Investigation:** Emma Thibault, Sean M. Mahoney, James V. Briskie, Matthew W. Reudink.

**Methodology:** Sean M. Mahoney, Matthew W. Reudink.

**Project administration:** Matthew W. Reudink.

**Supervision:** Matthew W. Reudink.

**Validation:** Matthew W. Reudink.

**Visualization:** Matthew W. Reudink.

**Writing – original draft:** Emma Thibault, James V. Briskie, Matthew W. Reudink.

**Writing – review & editing:** Emma Thibault, Sean M. Mahoney, James V. Briskie, Mateen Shaikh, Matthew W. Reudink.

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
