## [Decision Letter · Decision Letter 0]

13 Jul 2022

PONE-D-22-14733Extra-pair paternity drives plumage colour elaboration in male passerinesPLOS ONE

Dear Dr. Reudink,

Thank you for submitting your manuscript to PLOS ONE. After careful consideration, we feel that it has merit but does not fully meet PLOS ONE’s publication criteria as it currently stands. Therefore, we invite you to submit a revised version of the manuscript that addresses the points raised during the review process.

My apologies for how long this review took.  The second reviewer was unable to complete their review in a timely manner due to unforeseen events, so I have stepped in to act as the second reviewer myself.  I agree with reviewer one that this is a strong manuscript, and I look forward to your revision.  

We look forward to receiving your revised manuscript.

Kind regards,

Patrick R Stephens, Ph.D.

Academic Editor

PLOS ONE

Journal Requirements:

"Funding for this research was provided by an Natural Sciences and Engineering Research Council of Canada (NSERC) Discovery Grant #4603-2018 to MWR and a Thompson Rivers University CUEF UREAP award to ET."

Reviewers' comments:

Reviewer's Responses to Questions

**Comments to the Author**

1. Is the manuscript technically sound, and do the data support the conclusions?

Reviewer #1: Yes

2. Has the statistical analysis been performed appropriately and rigorously? 

Reviewer #1: Yes

3. Have the authors made all data underlying the findings in their manuscript fully available?

Reviewer #1: No

4. Is the manuscript presented in an intelligible fashion and written in standard English?

Reviewer #1: Yes

5. Review Comments to the Author

Reviewer #1: The authors address a fundamental hypothesis in avian sexual selection that has yet to be tested with a large and strong dataset. They make use of two publicly available databases and find convincing support for the idea that EPP rates correlate with male plumage elaboration and sexual dichromatism. The study is concise, direct, and strong. I have no issues with the manuscript framing, methods, or results. Well done.

6. PLOS authors have the option to publish the peer review history of their article (what does this mean?). If published, this will include your full peer review and any attached files.

Reviewer #1: No

**Reviewer two (associate editor): **

Overall this is a solid study on a topic of wide interest.  There some points of clarification needed in the methods, and assuming that your input trees were largely invariant for the 186 species of the study no new analyses will be needed.  However, if both the branch lengths and the topology vary, I believe you need to follow standard practice with the Jetz group trees and repeat your analysis on a sample of trees.  

Specific comments:

Male-like plumage patterns?

Line 112: please define “male like”.  Does it mean high variance among RBG scores from different parts of the body?    I assume that being more “male like” means more colorful or complex patterns, but I am not sure from the description here how exactly it is quantified.   Clarifications on choice of phylogeny.

You took a sample of 1000 trees and boiled them down to a single tree using maximum likelihood. This leads to several questions:

One http://birdtree.org?", Jetz and colleagues  suggest that all comparative analyses of birds be performed using sample of trees from their posterior distribution, and that is becoming standard practice when using the Jetz et al bird trees or the Upham et al. mammal trees.  Why did you do your analysis on a single tree rather than repeating it on a sample? Since you wanted to use a single tree, why didn’t you use the maximum clade credibility tree from Jetz et al.?  If there is a previous study that you can cite for the method that you used to create a ML tree from your sample of 1000, please mention it.  For the 1000 input trees the method you used should produce a nearly identical tree to the MCC tree, but you tree might differ somewhat from the MCC tree of the entire distribution (but see below about toplogy vs branch lengths). It looks like you used the Hackett backbone trees. How did you decide between the two potential backbone trees at birdtree.org?

*
Most importantly:
*

Did the topology of the trees you sampled vary? 

If the tree topology was stable for the 186 species in your study,  in my opinion the way you did the analyses is quite reasonable.  Small differences in the branch lengths of potential trees are highly unlikely to have affected the outcome of your analyses.  If this is the case I would add something to the methods like:

“For the 186 species included in our analyses the topology of the trees we sampled was invariant.  In order to derive branch lengths for the consensus tree used in our comparative analyses we . . . .”

I doubt even very particular readers will mind that you didn’t repeat the analysis 50+ times just to test the robustness of the results to potential branch length variation.  

If you aren’t sure about whether the topology varied, construct a strict consensus tree of the 1000 trees you sampled.  If the strict consensus is fully resolved, the topology does not vary among them.

**
*[If that is the case, ignore everything from here down!] *
**

However, if both the topology and branch lengths of the trees varied, I think you need to repeat your analyses on a sample of trees to assess the robustness of your results, as is quickly becoming standard practice. 

You don’t need to use all 1000 trees.  Nakegawa et al. (2019) suggest that 50-100 trees is generally sufficient for such an exercise.

Nakagawa S, De Villemereuil P. A General Method for Simultaneously Accounting for Phylogenetic and Species Sampling Uncertainty via Rubin’s Rules in Comparative Analysis. Syst Biol. 2019;68: 632–641. pmid:30597116

Further, even if you need to repeat your analyses, I would not attempt to incorporate a summary of the results across all those trees into the main MS in any detail.  The current manuscript is concise and well written, and making the main results harder to understand for the 5% of readers who are specialists in these methods does not seem warranted to me.  I think it’s perfectly fine to present results in the main paper using a single preferred tree.

For the replicate analyses, simply mention in the methods that you tested the robustness of your results across a sample of potential fully resolved trees from birdtree.org.  In the results report the % of trees in which you obtained qualitatively identical results.    

You can report results from the alternate trees in the supplementary materials however you like.  A table of model scores would be fine, or perhaps a histogram of correlations observed.  If the methods for conducting analyses on replicate trees would disrupt the flow of the current text (e.g., did you assume a lambda value of 1 or did you re-estimate it each time), you can also  describe how you did those in the supplement.

---

## [Author Response · Author response to Decision Letter 0]

4 Aug 2022

Please see attached response to reviewers document

---

## [Editor Report · Decision Letter 1]

8 Aug 2022

Extra-pair paternity drives plumage colour elaboration in male passerines

PONE-D-22-14733R1

Dear Dr. Reudink,

We’re pleased to inform you that your manuscript has been judged scientifically suitable for publication and will be formally accepted for publication once it meets all outstanding technical requirements.  

Kind regards,

Patrick R Stephens, Ph.D.

Academic Editor

PLOS ONE

---

## [Editor Report · Acceptance letter]

12 Aug 2022

PONE-D-22-14733R1 

Extra-pair paternity drives plumage colour elaboration in male passerines 

Dear Dr. Reudink:

I'm pleased to inform you that your manuscript has been deemed suitable for publication in PLOS ONE. Congratulations! Your manuscript is now with our production department. 

Kind regards, 

on behalf of

Dr. Patrick R Stephens 

Academic Editor

PLOS ONE